# Mechanisms of Arachidonic Acid In Vitro Tumoricidal Impact

**DOI:** 10.3390/molecules28041727

**Published:** 2023-02-11

**Authors:** Hatem Tallima, Rashika El Ridi

**Affiliations:** 1Department of Chemistry, School of Sciences and Engineering, The American University in Cairo, New Cairo 11835, Egypt; 2Zoology Department, Faculty of Science, Cairo University, Giza 12613, Egypt

**Keywords:** arachidonic acid, tumoricidal mechanisms, sphingomyelin, neutral sphingomyelinase, ceramide, β2 microglobulin, reactive oxygen species, caspases

## Abstract

To promote the potential of arachidonic acid (ARA) for cancer prevention and management, experiments were implemented to disclose the mechanisms of its tumoricidal action. Hepatocellular, lung, and breast carcinoma and normal hepatocytes cell lines were exposed to 0 or 50 μM ARA for 30 min and then assessed for proliferative capacity, surface membrane-associated sphingomyelin (SM) content, neutral sphingomyelinase (nSMase) activity, beta 2 microglobulin (β2 m) expression, and ceramide (Cer) levels. Reactive oxygen species (ROS) content and caspase 3/7 activity were evaluated. Exposure to ARA for 30 min led to impairment of the tumor cells’ proliferative capacity and revealed that the different cell lines display remarkably similar surface membrane SM content but diverse responses to ARA treatment. Arachidonic acid tumoricidal impact was shown to be associated with nSMase activation, exposure of cell surface membrane β2 m to antibody binding, and hydrolysis of SM to Cer, which accumulated on the cell surface and in the cytosol. The ARA and Cer-mediated inhibition of tumor cell viability appeared to be independent of ROS generation or caspase 3/7 activation. The data were compared and contrasted to findings reported in the literature on ARA tumoricidal mechanisms.

## 1. Introduction

There is a prevailing entrenched dogma advocating arachidonic acid (ARA) as a promoter of cancer initiation and spread [1], despite the fact that comprehensive literature searches found controversial and no firm, reliable, or highly significant associations between dietary ARA intake, blood or muscle ARA levels, and cancer risk [2,3,4,5,6,7,8,9]. Moreover, physiological free ARA levels (10–60 μg/mL) were reported to severely suppress the proliferation rate of human osteogenic sarcoma cells in culture [10]. At 15 μg/mL, ARA totally inhibited the proliferation of the human leukemic T cell line [11]. Proliferation and viability of human breast, lung, glioma, and prostate cancer cells were selectively inhibited following culture in medium supplemented with 20–30 μg/mL ARA, thus suggesting ARA possesses useful clinical potential in cancer treatment [12,13,14,15,16,17,18,19]. Additionally, exposure to 8 μM ARA for 48 h has been shown to reduce the proliferation and migration of breast cancer cell lines [19].

Several studies explored the mechanisms underlying ARA tumoricidal activity. Free, unesterified ARA, and not its eicosanoid metabolites, was found to have selective anti-tumor action and induce apoptosis of human retinoblastoma and glioma cells both in vitro and in vivo. The underlying mechanism of action involved, enhancing tumor cells’ generation of lipid peroxides, leads to the activation of caspase 3 and 8 and eventual apoptosis [13,14,18,19,20]. Exposure to >120 μM ARA for 48 h induced apoptosis of colon cancer cells via the generation of reactive oxygen species (ROS), activation of caspase 3 and caspase 9, and accumulation of intracellular calcium [21]. Conversely, treatment of breast and normal cancer cells with 8 μM ARA for 48 h selectively reduced tumor cell proliferation and viability via ROS generation and caspases activation, yet independently of calcium entry [19]. Lipid peroxidation was advocated as the major mechanism underlying ARA anti-cancer cell activity and appeared to be counteracted by host anti-oxidants, notably uric acid [13,14,20,21]. Conversely, exposure of HT-29 human colon cancer cells to 100 μM ARA for 2–4 days led to the death of 90% of the cells owing to the inhibition of the endogenous synthesis of fatty acids and induction of endoplasmic reticulum stress, independently of ROS, of which levels decreased following ARA exposure [22]. Changes in membrane fatty acid composition and the induction of ligand-activated transcription factors, PPARs (peroxisome proliferator-activated receptors) were implicated in the ARA (50 μM)-mediated death of the human lung, A-549 [23], and breast, MCF-7 [24] cell lines, associated with a decrease in caspase 3 and caspase 9 activity.

An entirely different mechanism underlying ARA anti-tumor potential has also been proposed. Clinically detectable cancers are characterized by a decline or lack of surface membrane expression of molecules critical to the interaction of immune cell effectors with tumor cells. Notably, cytotoxic CD8+ T lymphocytes kill tumor cells via the recognition and binding of tumor-associated antigens presented by the class I human leukocyte antigen (HLA-I) alpha chain and beta-2 microglobulin (β2 m). However, the complete downregulation of HLA-I and natural killer cell-activating molecules expression on the cell surface is common in tumor cells [25,26]. This preponderant immune evasion arm was ascribed to changes in the cell surface membrane biochemistry and organization, mainly involving the impairment of sphingomyelin (SM) content and function ([26] and references therein). A decrease in SM hydrolysis to phosphocholine and ceramide (Cer) would impair the cytosolic content of Cer, a major cell proliferation inhibitor and apoptosis inducer [26,27]. Cell surface membrane-associated neutral sphingomyelinase (nSMase) activity controls the hydrolysis of surface membrane SM and intracellular Cer content. Arachidonic acid is prominent among polyunsaturated fatty acids nSMase activators and, accordingly, was proposed for cancer therapy via its role in modulating cell surface membrane organization and increasing intracellular Cer levels [26,28].

Oral supplementation with ARA is not expected to lead to serum and tissues ARA concentrations higher than 50 μM for a brief period because albumin specifically binds to fatty acids. This rapidly reduces the concentration of free fatty acids to below 0.1 μM while permitting far higher concentrations, from 20 μM to >3 mM, to be stabilized in blood and tissues [29,30,31,32,33]. Of note, in vitro exposure to 50 μM ARA does not impair T cell viability [33]. Accordingly, we opted for 30 min exposure of tumor and normal cell lines to fetal calf serum-free medium supplemented with 50 μM ARA to investigate the ARA tumoricidal mechanism(s).

## 2. Results

### 2.1. Arachidonic Acid Impact on Cell Proliferation

The results of three independent experiments revealed that exogenous ARA significantly (*p* < 0.05) decreases the proliferation of all tumor cell lines, notably HepG2 cells, and affects the proliferative capacity of the normal liver hepatocyte cell line, BNL, only in wells with small cell numbers (Figure 1).

### 2.2. Cell Surface Membrane Sphingomyelin

#### 2.2.1. Reactivity to Lysenin in IF

Repeat experiments showed that except for the normal mouse hepatocytes, BNL, the lysenin indirect membrane immunofluorescence (IF) assay failed to visualize SM in all human tumor cell lines tested, whether intact or ARA (50 μM, 30 min, 37 °C)-treated. Conversely, surface membrane SM was consistently available to lysenin binding in BNL cells (Figure 2).

#### 2.2.2. Cell Surface Membrane Sphingomyelin Content

The cell surface membrane SM of intact HepG2 and MCF-7 differed from other cell lines in being more readily accessible (*p* < 0.05) to Triton X-100 extraction and sensitive to ARA and versene treatment, explaining higher values in intact compared to ARA-treated cells. Conversely, Huh-7 and BNL cells needed ARA impact on the surface membrane to allow efficient SM extraction by Triton X-100. The resilience of A-549 surface membrane SM extraction was not affected by ARA exposure (Figure 3).

### 2.3. Impact on Cell Surface Membrane Neutral Sphingomyelinase Activity

The data of two independent experiments illustrated in Figure 4 indicate that nSMase activity, as judged by delta mean fluorescence counts/well of 5 μg/well protein of surface membrane extracts of normal hepatocytes, BNL is significantly (*p* < 0.005) higher compared to tumor cells, and is unaffected by cell exposure to ARA for 30 min at 37 °C. Conversely, ARA exposure elicited a significant (*p* < 0.05) increase in nSMase activity in hepatic and lung cell carcinomas but not in breast cancer cells, MCF-7.

### 2.4. Exposure of Cell Surface Membrane β2 Microglobulin

All cells tested were entirely negative following incubation with anti-human β2 microglobulin (β2 m) antibody in direct IF except for BNL, the normal mouse hepatocytes. Treatment with 50 μM ARA for 30 min led to the weak, but definitely evident, exposure of tumor cells’ surface membrane β2 m to interaction with a specific monoclonal antibody (Figure 5).

### 2.5. Surface Membrane and Cytoplasmic Ceramide in Intact and Treated Cells

Arachidonic treatment of tumor cells led to the detection of otherwise poorly expressed surface membrane ceramide (Cer). Normal hepatocytes differed from the tumor cells in harboring readily detectable cytoplasmic Cer. The increase in cytoplasmic Cer expression in ARA-treated tumor cells was remarkable (Figure 6).

### 2.6. Reactive Oxygen Species in Cell Extracts

Repeat tests revealed that ARA treatment leads to a significant (*p* < 0.02) decrease in the ROS activity of the cell lines tested (Figure 7). Of note, intact MCF-7 cells showed the highest ROS levels, which decreased following ARA treatment in accordance with Bae et al. [22].

### 2.7. Caspase Activity

Repeat tests with identical results revealed that repeatedly passaged HepG2 cells likely lost caspase 3/7 expression or harbor an inhibitor of caspase 3/7 activity. Caspase 3/7 reactivity appeared to be inversely proportional to the malignancy of the cell lines, with normal mouse hepatocytes showing the highest and Huh-7 hepatic carcinoma cells showing higher content than A-549 and MCF-7 cells (Figure 8). Other striking results involved the inhibitory action (*p* < 0.02) of ARA exposure (50 μM, 30 min, 37 °C) on the cell lines’ caspase activity, as judged using the Caspase-Glow 3/7 Assay (Figure 8).

## 3. Discussion

Brief exposure to physiological ARA concentration elicited a significant decline in tumor cell proliferation and viability, in full accord with previous reports [10,11,12,13,14,15,16,17,18,19,20,21,22,23,24,25,26]. The impact on BNL cells, which are able to vigorously proliferate to confluence, is most likely owing to ARA’s role in promoting cell contact inhibition [26]. However, cancer and normal cell lines differed in many of their basic properties, notably the ease of extraction of their surface membrane SM. Numerous studies have documented the preponderant role of SM in cancer initiation, progression, and metastasis (reviewed in refs. [26,34,35,36,37]). Yet, few studies have examined the tumor cell surface membrane SM content and properties, despite the fact that tumor cell increased SM (d35:1) content was recently proposed as the most prominent and reliable predictor for lung adenocarcinoma recurrence after surgery [38]. Sphingomyelin (d18:0/d16:0) showed high expression in liver cancer metastatic lesions, thyroid papillary, breast, and gastric cancer, and low expression in prostate cancer [39]. Alteration in SM content was recently predicted to be fundamental in the differentiation between cancerous and normal tissue and likely between right and left colorectal cancers [40,41]. We could not find significant differences between SM content in the different cell lines, including the non-tumoral BNL. The differences rather involved the rigidity of the SM-based cloak. Cell surface membranes with SM absorbed in forming hydrogen bonds with neighboring molecules and surrounding water would easily be destabilized by exposure to Triton X-100, or ARA, followed by versene [42,43,44,45,46,47]. Surface membrane fluidity thus appeared higher in HepG2 and MCF-9 cells than in Huh-7 and normal hepatocytes [48,49,50]. Lung adenocarcinoma A-549 cells showed resistance to Triton X-100-mediated SM extraction before and after 50 μM/30 min ARA exposure. The SM membrane content of this cell line was also marginally affected following treatment with 100 μM resveratrol for 24 h [51].

The surface membrane of normal hepatocytes did not differ from tumor cells in SM content but in its unique accessibility to lysenin binding, irrespective of ARA exposure (Figure 1). It was consistently difficult to visualize tumor cell surface membrane SM via lysenin binding in indirect IF, even by spectral confocal microscopy [52]. Exposure to ARA failed to promote lysenin binding to tumor cells’ surface membrane SM, likely because of the significant activation of the surface membrane-associated nSMase, the enzyme responsible for the hydrolysis of surface membrane SM to Cer and phosphocholine. In support, Cer accumulated on the surface and cytosol of the different tumor cells. These findings together document the fundamental differences between the cell surface membranes of tumor and normal cells, supporting the view stating biochemical changes in outer cell membranes are critical in tumor initiation, progression, and metastasis [26]. Arachidonic acid-mediated powerful activation of nSMase overcame A-549 outer membrane rigidity and led to Cer accumulation at the cell surface and cytoplasm, a mechanism explaining ARA influence on A-549 cells proliferation and growth besides those previously proposed [23,24,53]. Nevertheless, tumor cell lines are not created equal, as breast adenocarcinoma MCF-7 cells displayed an ARA-dependent increase in the outer membrane and cytosolic Cer independently of nSMase activation. The difference could, however, be more apparent than real because of the restricted transcription of nSMase2 in MCF-7 cells [54,55,56].

Except for normal hepatocytes, the different cell lines’ surface membrane β2 m was unavailable to specific antibody binding in direct IF, in full accord with the numerous reports documenting the absent or low expression of MHC/HLA class I molecules on the tumor cell surface [25,26,57,58,59,60]. Exposure to ARA uniformly led to dim reactivity of otherwise concealed β2 m to the specific antibody, reflecting the more or less reversible alterations in tumor cells’ HLA class I expression [59,60]. Our results favor biochemical changes in the tumor cell surface to explain HLA-I downregulation or loss rather than a genetic modification or transcriptional alterations [25,60]. Even a slight increase in MHC class I and other molecules expression may render tumor cells susceptible to cytotoxic T and natural cell killing, a ground for recommending the nutrient ARA as safe and effective for tumor prophylaxis and therapy [26,61]. Destabilization of the tumor cell’s outer membrane following ARA exposure may well be the reason for exposing β2 M to antibody binding and suppressing tumor cell proliferation [62,63].

In addition to the potential of exposure to otherwise hidden surface membrane molecules that would be invaluable in in vivo settings [25,26,60], ARA elicited a remarkable accumulation of Cer at the cell surface and cytosol. Ceramide accumulation at both the cell surface [64,65] and the cytosol plays a preponderant role in mediating physiological and pharmacologically stimulated apoptosis, a ground for considering Cer a tumor-suppressor lipid [35,36,37,66]. Contrary to previous studies [13,14,18,19,20,21], the suppression of tumor cell proliferation and viability following ARA exposure and Cer accumulation may not be ascribed to the induction of ROS release or caspase 3/7 activation. However, our results concord with other studies reporting ARA-mediated tumor growth suppression independently of ROS release or caspase activation [16,22,23,24].

The controversy may be due to differences in ARA concentration and exposure time: 20 μM for 3 to 5 h [16], 8 μM continuously for 24 or 48 h [19], 1–50 μM for 24 h [24], and 100–150 μM for up to 72 h and more [20,21,22,23]. Differences in target tumor cell lines properties and characteristics are influential in directing and dictating ARA tumoricidal mechanisms. Caspase 3 is absent in MCF-7, but capase 7 is readily activated [24]. In accordance with our results, caspase 3 was recently shown to be weakly expressed in intact hepatocellular carcinoma HepG2 cells [67], supporting earlier findings documenting the lower expression of caspase 1 and caspase 3 in hepatocellular carcinoma tissue compared to nontumor cells [68,69]. Conversely, caspase 3 was found to be overexpressed in HepG2 and absent in colon carcinoma cell lines [70]. Of note, caspase 3/7 activity in liver cancer cellsHuh-7 cells and normal hepatocytes, BNL appeared nearly similar in support with the statement correlating caspase 3 expression with hepatocellular carcinoma differentiation [69,71]. Finally, nSMase activation and Cer accumulation may mediate cell growth arrest and apoptosis independently of inducting ROS storm and caspases activation via numerous extracellular and intracellular mechanisms [72,73,74,75,76,77,78,79,80,81].

The results confirmed ARA’s potential tumoricidal role [10,11,12,13,14,15,16,17,18,19,20,21,22,23,24,25,26,27,28]. Being a sorely needed nutrient and essential component of cell membranes are grounds for advocating the safety of ARA administration, which has otherwise been fully documented [1,2,3,4,5,6,7,8,9]. The mechanisms of action appear mild, independent of ROS induction and caspase activation, essentially involving cell surface membrane destabilization and nSMase activation. Changes in tumor surface membrane organization may restore contact inhibition, the hallmark of normal versus cancer cells, and explains ARA’s impact on tumor and non-cancerous cells’ proliferative capacity. Additionally, ARA-dependent nSMase activation with subsequent surface membrane SM replacement by Cer allows exposure of hitherto concealed molecules for in vivo life-saving interactions with the host immune system effectors. Cytosolic Cer accumulation in the cytosol leads to tumor cell growth arrest. These findings fully support our advocacy involving concentration on the importance of the cell surface in cancer initiation and progression and the potential of ARA administration for cancer prophylaxis and therapy.

## 4. Materials and Methods

### 4.1. Tumor and Normal Cells

Cell lines of hepatocellular carcinoma, HepG2, liver cancer, Huh-7, human nonsmall lung adenocarcinoma, A-549, breast adenocarcinoma, MCF-7, and normal mouse liver BNL were obtained from Nawah Scientific Egypt (Al Mokattam, Cairo, Egypt) and maintained at 37 °C/5% CO_2_ in Dulbecco’s modified Eagle medium/4.5 g/L glucose with L-glutamine and sodium pyruvate (DMEM) supplemented with 100 Units/mL penicillin, 100 μg/mL streptomycin, and 10% heat-inactivated (56 °C, 30 min) fetal calf serum (Lonza Bioscience, Verviers, Belgium). Lonza Trypsin-Versene^TM^ was used for cell passage.

### 4.2. Arachidonic Acid Treatment

Adherent cells were exposed to 50 μM (15 μg/mL) ARA (Cayman Chemical, Neratovice, Czech Republic) in fetal calf serum (FCS)-free Dulbecco’s phosphate-buffered saline, pH 7.1 (D-PBS) for 30 min, at 37 °C and no CO_2_, and then washed subsequently in DMEM/FCS and/or FCS-free medium to remove all ARA traces.

### 4.3. Arachidonic Acid Impact on Cell Proliferation

Serial dilutions of 200,000 cells were left to adhere overnight at 37 °C/5% CO_2_ in duplicate wells of flat-bottomed sterile tissue culture plates, washed with FCS-free D-PBS before exposure to 0 or 50 μM ARA at 37 °C for 30 min. Cells were washed 3× with 250 μL/well DMEM/FCS to remove all ARA traces and left to recover at 37 °C/5% CO_2_ for 4 h before adding all wells of two parallel plates with 20% Cell Titer 96 AQ_ueous_ One Solution Cell Proliferation Reagent (Promega, Madison, WI, USA) [82] or alamarBlue Cell Viability Reagent (ThermoFisher Scientific, Waltham, MA, USA) [83]. After overnight incubation to allow the cells to vigorously proliferate, absorbance was read at 492 nm ((Multiskan EX, Labsystems, Helsinki, Finland), and fluorescence at Excitation (Ex) 540/Emission (Em) 590 (Victor X4 Multi-Label Plate Reader, PerkinElmer, Waltham, MA, USA) for Cell Titer and Alamar Blue reagents, respectively.

### 4.4. Preparation of Cell Surface Membrane Extracts

Adherent cells in parallel 75 mL flasks were washed with sterile D-PBS, exposed to 0 or 50 μM ARA in D-PBS at 37 °C for 30 min, washed in D-PBS, then dislodged with trypsin-versene and retrieved cells repeatedly washed in D-PBS to remove all ARA and trypsin-versene traces. Twenty million intact and ARA-treated cells were incubated for 30 min at room temperature [46] with 1:3 volumes D-PBS supplemented with 0.1% Triton X-100 (Promega) and protease inhibitors (4 μg/mL leupeptin and 2 mM phenyl methyl sulfonyl fluoride, from Sigma-Merck). After brief vortexing, the supernatants containing cell surface membrane Triton-soluble and Triton-insoluble proteins and lipids were assessed for protein content, aliquoted, and stored at −20 °C until use for evaluation of SM content and surface membrane associated-nSMase activity, and thawed only once.

### 4.5. Preparation of Cell Homogenates Extracts

Cells remaining following extraction of surface membrane molecules were thoroughly homogenized for 30 min in D-PBS on ice, centrifuged at 400× *g*, and the supernatant retrieved in ice-cold reaction tubes, assessed for protein content as described above and stored at −20 °C until use for evaluation of reactive oxygen species (ROS) and caspase activity.

### 4.6. Cell Surface Membrane Sphingomyelin Content

#### 4.6.1. Sphingomyelin Detection

Expression of surface membrane SM was evaluated in highly viable untreated and ARA-treated cells via indirect membrane immunofluorescence (IF) probing using lysenin (41 kDa), an SM-specific binding protein from the earthworm *Eisenia foetida* [84,85]. Paraformaldehyde-fixed adherent cells in wells of flat-bottomed tissue culture microplates were exposed to 0 or 3.0 μg/mL lysenin (Sigma) at room temperature for 2 h, washed in DMEM/5% FCS, and then incubated for 1 h at room temperature in the presence of 1:100-diluted anti-lysenin rabbit antibodies (MyBioSource, San Diego, CA, USA). After washing, cells were incubated with 1:100-diluted, fluorescein isothiocyanate (FITC)-labeled goat anti-rabbit immunoglobulin conjugate (Sigma) and inspected by alternate light and ultraviolet Microscopy (Olympus Inverted Microscope, Tokyo, Japan).

#### 4.6.2. Sphinomyelin Content

Sphingomyelin content was measured in surface membrane Triton X-100 extracts of untreated and ARA-treated cells (5 μg protein in duplicate wells) using Sphingomyelin Assay Kit (Colorimetric), ab287856, (Abcam, Cambridge Biomedical Campus, Cambridge, UK), following the manufacturer’s steps. Absorption of duplicate wells was evaluated at 570 nm.

### 4.7. Neutral Sphingomyelinase Activity

Duplicate aliquots of 5 μg surface membrane extract proteins of untreated and ARA (50 μM, 30 min)-treated cells were evaluated for nSMase content using the Sphingomyelinase Assay kit of Abcam, ab287874 following the manufacturer’s instruction. Fluorescence was measured at Ex/Em of 540/590 nm, at 30, 60, and 120 min after adding reagents, and data were displayed after subtracting background values (Victor X4 Multi-Label Plate Reader).

### 4.8. Exposure of Cell Surface Membrane Beta 2 Microglobulin

Adherent cells in wells of flat-bottomed tissue culture microplates were exposed to 0 or 50 μM ARA, washed 3× with DMEM/FCS, fixed in 4% paraformaldehyde for 10 min, and washed again before incubation with 1:50-diluted FITC-labeled mouse monoclonal antibody to human β2 microglobulin (β2 m) (BioLegend, San Diego, CA, USA) for 1 h at room temperature. Cells in duplicate wells were examined and photographed under light and ultraviolet microscopy (Olympus).

### 4.9. Cell Surface Membrane and Cytoplasmic Ceramide

Untreated or ARA-treated, paraformaldehyde-fixed adherent cells in wells of flat-bottomed tissue culture microplates were repeatedly washed before and after brief exposure to D-PBS supplemented with 0 (surface membrane ceramide) or 0.1% (cytoplasmic ceramide) Triton X-100, and cells were then incubated for 1 h at room temperature in the presence of 1:20-diluted mouse monoclonal antibody to ceramide (Sigma- Aldrich- Merck). After washing in DMEM/1% FCS, cells were incubated for 1 h at room temperature with 1:50-diluted FITC-labeled antibody to mouse IgG (H+L) [F(ab’)_2_ fragment of affinity isolated goat antibody, adsorbed with bovine, horse, and human serum proteins (Sigma)]. Cells in duplicate wells were examined and photographed under light and ultraviolet microscopy (Olympus).

### 4.10. Reactive Oxygen Species Content in Cell Homogenate Extracts

Aliquots of 5 and 10 μg of proteins of intact and ARA-treated cell homogenate extracts were incubated for 1 h at room temperature in the dark with 20 μM 2′,7′-dichlorodihydrofluorescein diacetate (DCHF-DA) [21,86,87,88]. Reactive oxygen species release was estimated by fluorescence spectroscopy at Ex/Em of 485/535 nm, and data were displayed after subtracting background values.

### 4.11. Caspase Activity

Caspase 3/7 activity in 5 μg proteins of intact and ARA-treated cell homogenate extracts was evaluated using the Caspase-Glo 3/7 Assay of Promega, following the manufacturer’s instructions, whereby released luminescence minus background values is proportional to the amount of caspase activity present [19,67,89,90].

### 4.12. Statistical Analyses

All values were tested for normality. The Student’s two-tailed paired t-test and one-way ANOVA with post-test were used to analyze the statistical significance of differences between selected values and considered significant at *p* < 0.05 (GraphPad InStat, San Diego, CA, USA).

## 5. Conclusions

Arachidonic is a structural constituent of cell membranes and necessary nutrient found in milk, eggs, meat, and fish. Intake of ARA does not lead to the production of inflammatory metabolites [91,92], which are generated as a result of cell stress or injury and pathogen invasion [61]. Exogenous ARA induces changes in the cell surface membrane organization, likely leading to exposure of molecules necessary for proper contact inhibition [93,94,95,96] or interaction with host immune effectors. Arachidonic acid activation of cell surface membrane-associated nSMase elicits accumulation of Cer, which is instrumental in the suppression of undue cell proliferation. Accordingly, ARA may be considered for safe prophylactic and therapeutic tumor treatment, contingent on the assessment of its efficacy and mechanisms of action against the growth of cancer cell lines in the nude mouse model.

## Figures and Tables

**Figure 1 molecules-28-01727-f001:**
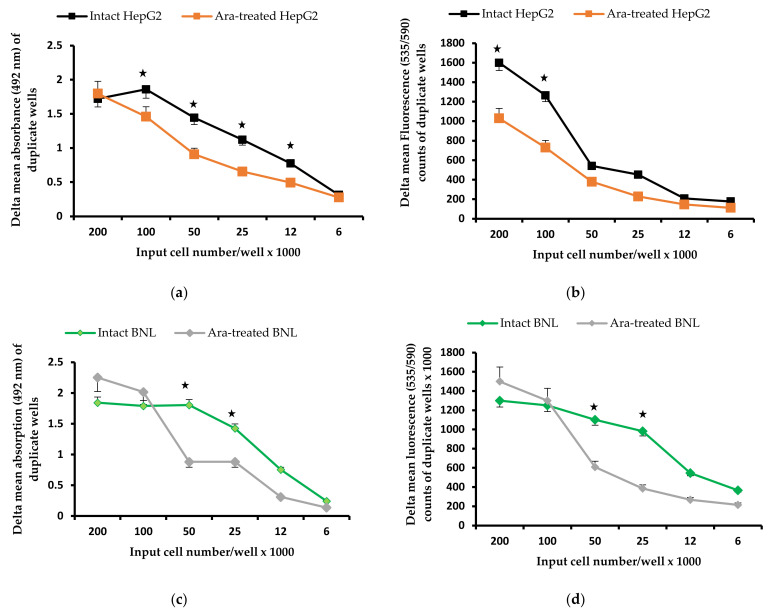
Impact of ARA treatment on cell proliferation. Representative of three experiments. Untreated and ARA-treated hepatic cell carcinoma, HepG2 (**a**,**b**) and normal hepatocytes, BNL (**c**,**d**) proliferative capacity was assessed by colorimetric (Cell Titer 96 AQ_ueous_ One Solution Cell Proliferation Reagent, (**a**,**c**)), and fluorometric (alamarBlue Cell Viability Reagent, (**b**,**d**)) assays. Delta = background absorption or fluorescence values subtracted. Values recorded in 3 independent experiments were statistically analyzed, and differences *(p* < 0.05) were marked with an asterisk.

**Figure 2 molecules-28-01727-f002:**
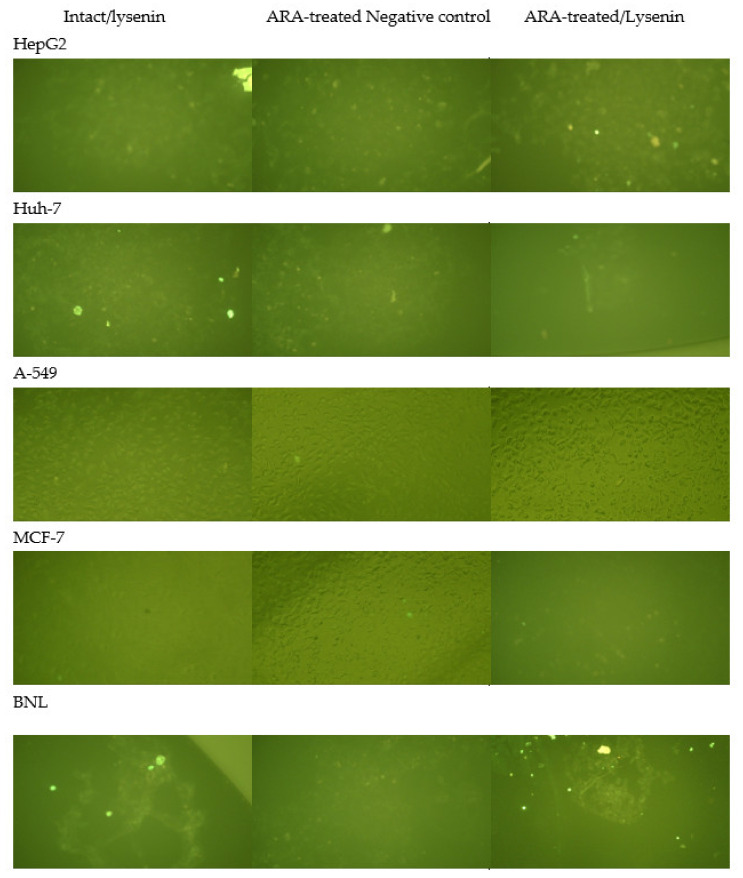
Reactivity of tumor and normal cell surface membrane to lysenin in indirect membrane immunofluorescence. Intact and ARA-treated cells were reacted with 0 (negative control) or 3 μg/mL lysenin in indirect membrane IF. X200.

**Figure 3 molecules-28-01727-f003:**
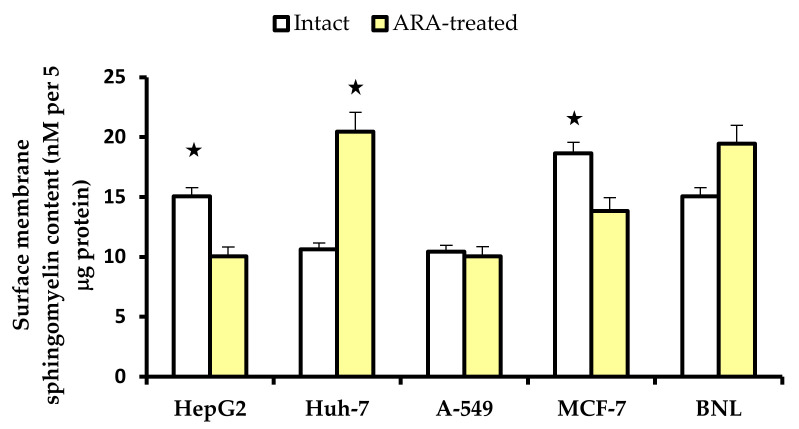
Content of Triton X-100-extracted surface membrane sphingomyelin of intact and arachidonic acid (ARA)-treated cells. Each column represents the mean SM content in 5 μg protein assessed in 2 independent experiments, and vertical bars denote the standard deviation around the mean. Triton X-100-extracted SM content was significantly (*p* < 0.05, asterik) higher in intact versus ARA-treated HepG2 and MCF-7 cells and ARA-treated compared to intact Huh-7 cells.

**Figure 4 molecules-28-01727-f004:**
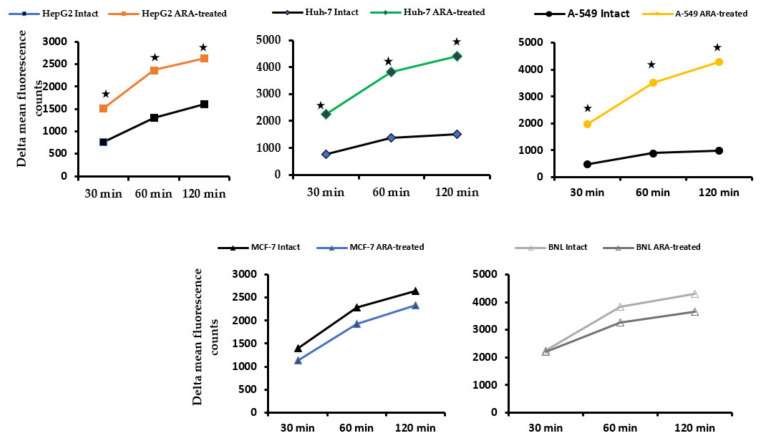
Kinetics of surface membrane-associated neutral sphingomyelinase activity. Surface membrane Triton X-100 extracts (5 μg protein/well) were assayed in duplicate wells. Each point represents delta (background fluorescence counts-subtracted) mean fluorescence counts at 30, 60, and 120 min after adding the probe of two independent experiments. Standard deviation values were uniformly less than 5%. Asterisks indicate statistical differences (*p* < 0.05) between ARA-treated and intact cells.

**Figure 5 molecules-28-01727-f005:**
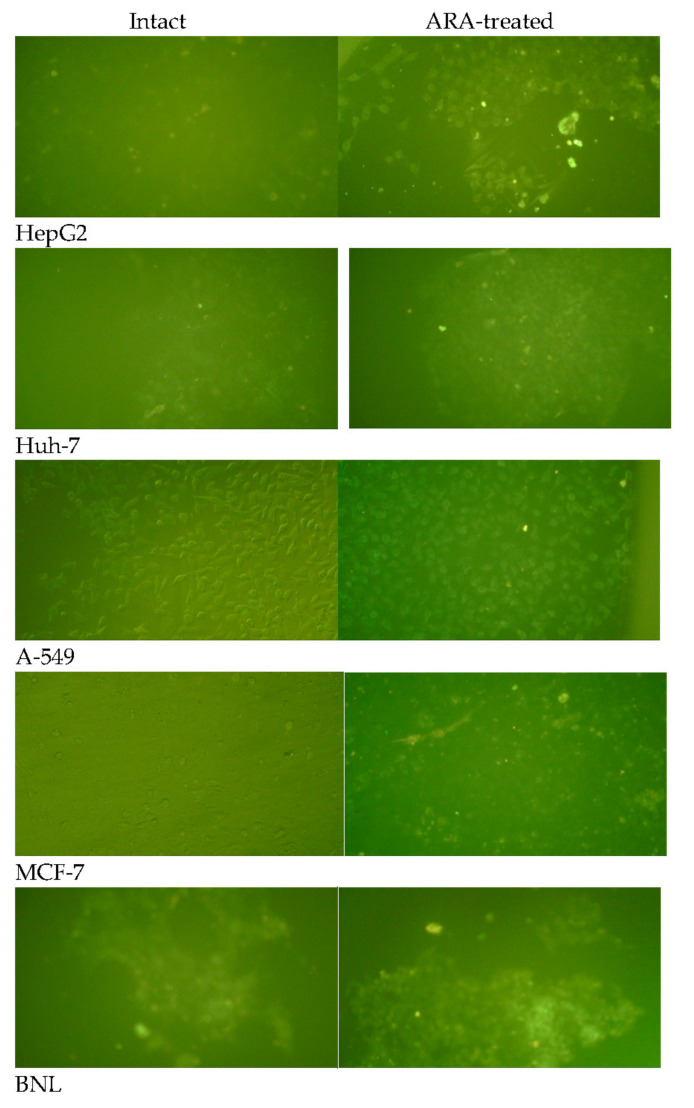
Surface membrane beta 2 microglobulin expression. Adherent cells were exposed to 0 (intact) or 50 μM arachidonic acid (ARA-treated) for 30 min and then tested for reactivity to anti-human β2 m antibody in direct membrane IF. X200.

**Figure 6 molecules-28-01727-f006:**
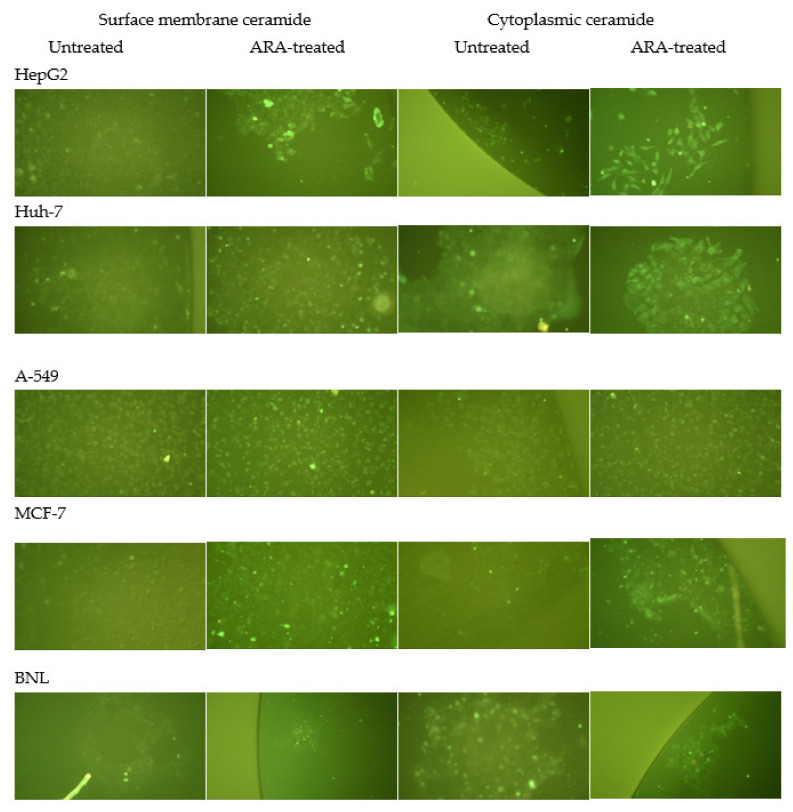
Expression of surface membrane and cytoplasmic ceramide in intact and ARA-treated cell lines. Adherent cells briefly exposed to 0 (surface membrane) or 0.1% (cytoplasmic) Triton-X 100 were incubated with 0 (untreated) or 50 μM arachidonic acid (ARA-treated) for 30 min and then tested for reactivity to anti-ceramide antibody in indirect IF. X200.

**Figure 7 molecules-28-01727-f007:**
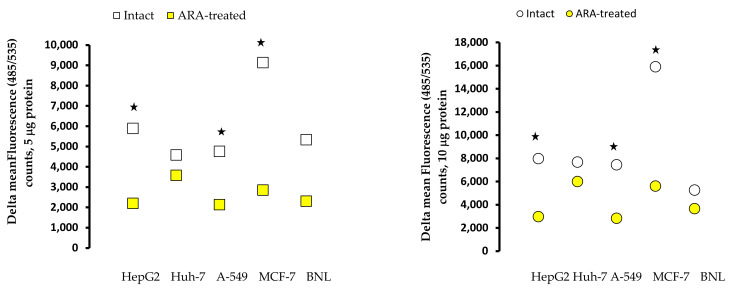
Reactive oxygen species in 5 and 10 μg cell homogenates extract protein. Each point represents delta (background fluorescence counts-subtracted) mean fluorescence counts recorded in 2 duplicate wells of two independent experiments with a standard deviation of less than 5%. Asterisks indicate statistical differences (*p* < 0.05) between intact and ARA-treated cells.

**Figure 8 molecules-28-01727-f008:**
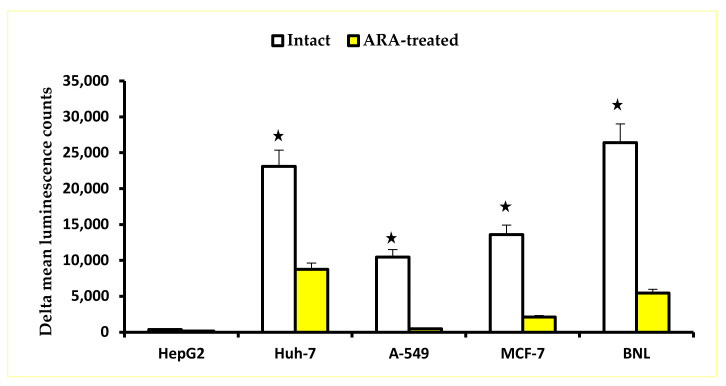
Caspase 3/7 reactivity in 5 μg cell homogenates extracts protein. Each column represents delta (background luminescence counts-subtracted) mean luminescence counts recorded in 2 duplicate wells of two independent experiments, and vertical bars depict the standard deviation around the mean. Asterisks indicate statistical differences (*p* < 0.02) between intact and ARA-treated cells.

## Data Availability

All data supporting reported results have been shown in the manuscript.

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
