# Peer review of "Mechanisms of Arachidonic Acid In Vitro Tumoricidal Impact"

_molecules, 2023, doi:10.3390/molecules28041727_

Round 1
Reviewer 1 Report
The research conducted by Tallima and Ridi is completely focused on showcasing the anticancer effect of Arachidonic Acid through the in vitro studies conducted on Hepatocellular, lung, and breast carcinoma and normal hepatocytes cell lines. The study looks novel, however, their are certain modifications that are required prior to its publication.
Abstract: Please focus on key findings
Language: Need to be significantly improved.
Introduction: Needs substantial revision. It is too short. The background/rationale of the study is missing. Write at least one paragraph on that with support of prior literature.
Results and discussions: Focus on your obtained data and prove its superiority over the existing research to make the readers understand the importance of this new study.
Conclusion: Provide some recommendation related to its positive outcomes with future pre-clinical studies that can be explored on suitable animal model.
Author Response
Open Review-1
( ) I would not like to sign my review report
(x) I would like to sign my review report
English language and style
( ) English very difficult to understand/incomprehensible
( ) Extensive editing of English language and style required
(x) Moderate English changes required
( ) English language and style are fine/minor spell check required
( ) I don't feel qualified to judge about the English language and style
|
Yes |
Can be improved |
Must be improved |
Not applicable |
|
|
Does the introduction provide sufficient background and include all relevant references? |
( ) |
( ) |
(x) |
( ) |
|
Are all the cited references relevant to the research? |
(x) |
( ) |
( ) |
( ) |
|
Is the research design appropriate? |
(x) |
( ) |
( ) |
( ) |
|
Are the methods adequately described? |
(x) |
( ) |
( ) |
( ) |
|
Are the results clearly presented? |
( ) |
( ) |
(x) |
( ) |
|
Are the conclusions supported by the results? |
( ) |
(x) |
( ) |
( ) |
Comments and Suggestions for Authors
The research conducted by Tallima and Ridi is completely focused on showcasing the anticancer effect of Arachidonic Acid through the in vitro studies conducted on Hepatocellular, lung, and breast carcinoma and normal hepatocytes cell lines. The study looks novel, however, there are certain modifications that are required prior to its publication.
Abstract: Please focus on key findings
Reply: The last sentence, which does not relate to key findings was deleted, and replaced by a sentence relevant to the Discussion.
Language: Need to be significantly improved.
Reply: The text was thoroughly revised by a native language speaker.
Introduction: Needs substantial revision. It is too short. The background/rationale of the study is missing. Write at least one paragraph on that with support of prior literature.
Reply: We thank and hail the eminent Reviewer for this excellent remark. The background/rationale of the study was indeed missing from fear of self-plagiarism from reference # 28 (now #26). The missing paragraph and novel references were added, and changes red marked.
Results and discussions: Focus on your obtained data
Reply: Therefore, the sentences: "supports the contention it represents a major mechanism underlying ARA tumoricidal potential", in Results 2.4., and "indicating that induction of ROS generation is not a mechanism underlying ARA tumoricidal impact", in Results 2.6 were deleted
and prove its superiority over the existing research to make the readers understand the importance of this new study.
Reply: Every effort was made towards his end, and additions red marked.
Conclusion: Provide some recommendation related to its positive outcomes with future pre-clinical studies that can be explored on suitable animal model.
Reply: Done with thanks and additions red marked.
Reviewer 2 Report
Authors have done a great effort in this manuscript, describing effects of arachidonic acid on different tumour cell lines.
However, some improvements must be done to consider this manuscript for publishing.
Result section and all the figure legends are required to be more descriptive, and number of experiments, replicates and statistics have to be included in and legends.
Please also include statistics to graphs (i.e. asterisks)
Fluorescence background must be subtracted from the images shown in figures 4 and 5, in order to better demonstrate what authors conclude.
Figure 1. It is concluded that ARA decreases viability more in tumour cells than in normal cells. Not sure if this reviewer is misunderstanding graphs but ARA decreases viability in both cell types.
Figures 3 and 6. There are not error bars, not statistics shown. What happens with fluorescence at time 0 in figure 3?
Figure S1 should be a main figure.
Material and Methods number assignations should be review. It seems order was changed but numbers were not (4.5 after 4.3; 4.4 after 4.9)
Author Response
Open Review-2
(x) I would not like to sign my review report
( ) I would like to sign my review report
English language and style
( ) English very difficult to understand/incomprehensible
( ) Extensive editing of English language and style required
( ) Moderate English changes required
( ) English language and style are fine/minor spell check required
(x) I don't feel qualified to judge about the English language and style
|
Yes |
Can be improved |
Must be improved |
Not applicable |
|
|
Does the introduction provide sufficient background and include all relevant references? |
(x) |
( ) |
( ) |
( ) |
|
Are all the cited references relevant to the research? |
( ) |
( ) |
( ) |
(x) |
|
Is the research design appropriate? |
(x) |
( ) |
( ) |
( ) |
|
Are the methods adequately described? |
(x) |
( ) |
( ) |
( ) |
|
Are the results clearly presented? |
( ) |
( ) |
(x) |
( ) |
|
Are the conclusions supported by the results? |
( ) |
(x) |
( ) |
( ) |
Comments and Suggestions for Authors
Authors have done a great effort in this manuscript, describing effects of arachidonic acid on different tumour cell lines.
Reply: Thank you very much.
However, some improvements must be done to consider this manuscript for publishing.
Reply: All done with heartfelt thanks, and changes red marked.
Result section and all the figure legends are required to be more descriptive, and number of experiments, replicates and statistics have to be included in and legends.
Reply: Thanks are due for this important criticism. Figure legends are now more descriptive, notably concerning number of experiments, replicates and statistics, and additions red marked.
Please also include statistics to graphs (i.e. asterisks).
Reply: Done for each relevant Figure.
Fluorescence background must be subtracted from the images shown in figures 4 and 5, in order to better demonstrate what authors conclude.
Reply: Fluorescence background was always subtracted before display of findings and that information is now added in the Materials and Methods section and Legends to Figures and red marked.
Figure 1. It is concluded that ARA decreases viability more in tumour cells than in normal cells. Not sure if this reviewer is misunderstanding graphs but ARA decreases viability in both cell types.
Reply: ARA impact may be explained by the fact that BNL cells actively divide to confluence, differently from T cells, which were shown to be unaffected by exposure to 50 µM ARA [33]. This issue was addressed, with thanks, in the Discussion, second sentence of the first paragraph, and third sentence in the last paragraph, both red marked.
Figures 3 and 6. There are not error bars, not statistics shown. What happens with fluorescence at time 0 in figure 3?
Reply: Thanks for this important comment. Error bars and statistics are now shown in legends to the Figures. Background fluorescence (fluorescence at time zero) is measured and subtracted at each time interval in wells containing cells, buffers, but no probe.
Figure S1 should be a main figure.
Reply: Done as recommended
Material and Methods number assignations should be review. It seems order was changed but numbers were not (4.5 after 4.3; 4.4 after 4.9).
Reply: Done with thanks and apologies.
Round 2
Reviewer 1 Report
The authors have substantially revised their manuscript. It can be accepted in its present form.
Reviewer 2 Report
None